# Direct Estimation of Differential Functional Graphical Models

**Boxin Zhao**
Department of Statistics
The Unveristy of Chicago
Chicago, IL 60637
boxinz@uchicago.edu

**Y. Samuel Wang**
Booth School of Business
The Unveristy of Chicago
Chicago, IL 60637
swang24@uchicago.edu

**Mladen Kolar**
Booth School of Business
The Unveristy of Chicago
Chicago, IL 60637
mkolar@chicagobooth.edu

## Abstract

We consider the problem of estimating the difference between two functional undirected graphical models with shared structures. In many applications, data are naturally regarded as high-dimensional random function vectors rather than multivariate scalars. For example, electroencephalography (EEG) data are more appropriately treated as functions of time. In these problems, not only can the number of functions measured per sample be large, but each function is itself an infinite dimensional object, making estimation of model parameters challenging. We develop a method that directly estimates the difference of graphs, avoiding separate estimation of each graph, and show it is consistent in certain high-dimensional settings. We illustrate finite sample properties of our method through simulation studies. Finally, we apply our method to EEG data to uncover differences in functional brain connectivity between alcoholics and control subjects.

## 1 Introduction

Undirected graphical models are widely used to compactly represent pairwise conditional independence in complex systems. Let $G = \{V, E\}$ denote an undirected graph where $V$ is the set of vertices with $|V| = p$ and $E \subset V^2$ is the set of edges. For a random vector $X = (X_1, \ldots, X_p)^T$, we say that $X$ satisfies the pairwise Markov property with respect to $G$ if $X_v \not\perp\!\!\!\perp X_w | \{X_u\}_{u \in V \setminus \{v,w\}}$ implies $\{v, w\} \in E$. When $X$ follows a multivariate Gaussian distribution with covariance $\Sigma = \Theta^{-1}$, then $\Theta_{vw} \neq 0$ implies $\{v, w\} \in E$. Thus, recovering the structure of the undirected graph is equivalent to estimating the support of the precision matrix, $\Theta$ [10, 13, 4, 24, 25].

We consider a setting where we observe two samples $X$ and $Y$ from (possibly) different distributions, and the primary object of interest is the difference between the conditional dependencies of each population rather than the conditional dependencies in each population. For example, in Section 4.3 we analyze neuroscience data sampled from a control group and a group of alcoholics, and seek to understand how the brain functional connectivity patterns in the alcoholics differ from the control group. Thus, in this paper, the object of interest is the *differential graph*, $G_\Delta = \{V, E_\Delta\}$, which is defined as the difference between the precision matrix of $X$, $\Theta^X$ and the precision matrix of $Y$, $\Theta^Y$, $\Delta = \Theta^X - \Theta^Y$. When $\Delta_{vw} \neq 0$, it implies $\{v, w\} \in E_\Delta$. This type of differential model has been adopted in [30, 22, 3].

In this paper, we are interested in estimating the differential graph in a more complicated setting. Instead of observing vector valued data, we assume the data are actually random vector valued functions (see [5] for a detailed exposition of random functions). Indeed, we aim to estimate the difference between two functional graphical models and the method we propose combines ideas from graphical models for functional data and direct estimation of differential graphs.

Multivariate observations measured across time can be modeled as arising from distinct, but similar distributions [9]. However, in some cases, it may be more natural to assume the data are measurements of an underlying continuous process [31, 18, 11, 28]. [31, 18] treat data as curves distributed according to a *multivariate Gaussian process* (MGP). [31] shows that Markov properties hold for Gaussian processes, while [18] shows how to consistently estimate underlying conditional independencies.

We adopt the functional data point of view and assume the data are curves distributed according to a MGP. However, we consider two samples from distinct populations with the primary goal of characterizing the difference between the conditional cross-covariance functions of each population. Naively, one could apply the procedure of [18] to each sample, and then directly compare the resulting estimated conditional independence structures. However, this approach would require sparsity in both of the underlying conditional independence graphs and would preclude many practical cases; e.g., neither graph could contain hub-nodes with large degree. We develop a novel procedure that directly learns the difference between the conditional independence structures underlying two MGPs. Under an assumption that the difference is sparse, we can consistently learn the structure of the differential graph, even in the setting where individual graphs are dense and separate estimation would suffer.

Our paper builds on recent literature on graphical models for vector valued data, which suggests that direct estimation of the differences between parameters of underlying distributions may yield better results. [12] considers data arising from pairwise interaction exponential families and propose the *Kullback-Leibler Importance Estimation Procedure* (KLIEP) to explicitly estimate the ratio of densities. [21] uses KLIEP as a first step to directly estimate the difference between two directed graphs. Alternatively, [30, 26] consider two multivariate Gaussian samples, and directly estimate the difference between the two precision matrices. When the difference is sparse, it can be consistently estimated even in the high-dimensional setting with dense underlying precision matrices. [22] extends this approach to Gaussian copula models.

The rest of the paper is organized as follows. In Section 2 we introduce our method for **Fu**nctional **D**ifferential **G**raph **E**stimation (FuDGE). In Section 3 we provide conditions under which FuDGE consistently recovers the true differential graph. Simulations and real data analysis are provided in Section 4[1]. Discussion is provided in Section 5. Appendix contains all the technical proofs and additional simulation results.

We briefly introduce some notation used throughout the rest of the paper. Let $|\cdot|_p$ denote vector $p$-norm and $\|\cdot\|_p$ denote the matrix/operator $p$-norm. For example, for a $p \times p$ matrix $A$ with entries $a_{jk}$, $|A|_1 = \sum_{j,k} |a_{jk}|$, $\|A\|_1 = \max_k \sum_j |a_{jk}|$, $|A|_\infty = \max_{j,k} |a_{jk}|$, and $\|A\|_\infty = \max_j \sum_k |a_{jk}|$. Let $a_n \asymp b_n$ denote that $z_1 \le \inf_n |a_n/b_n| \le \sup_n |a_n/b_n| \le z_2$ for some positive constants $z_1$ and $z_2$. Let $\lambda_{\min}(A)$ and $\lambda_{\max}(A)$ denote the minimum and maximum eigenvalues, respectively. For a bivariate function $g(s,t)$, we define the Hilbert-Schmidt norm of $g(s,t)$ (or equivalently, the norm of the integral operator it corresponds to) as $\|g\|_{\mathrm{HS}}^2 = \int \int \{g(s,t)\}^2 ds dt$.

## 2 Methodology

### 2.1 Functional differential graphical model

Let $X_i(t) = (X_{i1}(t), \ldots, X_{ip}(t))^T$, $i = 1, \ldots, n_X$, and $Y_i(t) = (Y_{i1}(t), \ldots, Y_{ip}(t))^T$, $i = 1, \ldots, n_Y$, be iid $p$-dimensional *multivariate Gaussian processes* with mean zero and common domain $\mathcal{T}$ from two different, but connected population distributions, where $\mathcal{T}$ is a closed subset of the real line.[2] Also, assume that for $j = 1, \ldots, p$, $X_{ij}(t)$ and $Y_{ij}(t)$ are random elements of a separable Hilbert space $\mathbb{H}$. For brevity, we will generally only explicitly define notation for $X_i(t)$; however, the reader should note that all notations for $Y_i(t)$ are defined analogously.

Following [18], we define the conditional cross-covariance function for $X_i(t)$ as

$$C_{jl}^X(s,t) = \mathrm{Cov}\left(X_{ij}(s), X_{il}(t) \mid \{X_{ik}(\cdot)\}_{k \neq j,l}\right). \tag{2.1}$$

If $C_{jl}^X(s,t) = 0$ for all $s, t \in \mathcal{T}$, then the random functions $X_{ij}(t)$ and $X_{il}(t)$ are conditionally independent given the other random functions. The graph $G_X = \{V, E_X\}$ represents the pairwise

Markov properties of $X_i(t)$ if

$$E_X = \{(j,l) \in V^2 : j \neq l \text{ and } \exists \{s,t\} \in \mathcal{T}^2 \text{ such that } C_{jl}^X(s,t) \neq 0\}. \quad (2.2)$$

In this paper, the object of interest is $C^\Delta(s,t)$ where $C_{jl}^\Delta(s,t) = C_{jl}^X(s,t) - C_{jl}^Y(s,t)$. We define the differential graph to be $G_\Delta = \{V, E_\Delta\}$, where

$$E_\Delta = \{(j,l) \in V^2 : j \neq l \text{ and } \|C_{jl}^\Delta\|_{HS} \neq 0\}. \quad (2.3)$$

Again, we include an edge between $j$ and $l$, if the conditional dependence between $X_{ij}(t)$ and $X_{il}(t)$ given all the other curves differs from that of $Y_{ij}(t)$ and $Y_{il}(t)$ given all the other curves.

## 2.2 Functional principal component analysis

Since $X_i(t)$ and $Y_i(t)$ are infinite dimensional objects, for practical estimation, we reduce the dimensionality using *functional principal component analysis* (FPCA). Similar to the way principal component analysis provides an $L_2$ optimal lower dimensional representation of vector valued data, FPCA provides an $L_2$ optimal finite dimensional representation of functional data. As in [18], for simplicity of exposition, we assume that we fully observe the functions $X_i(t)$ and $Y_i(t)$. However, FPCA can also be applied to both densely and sparsely observed functional data, as well as data containing measurement errors. Such an extension is straightforward, cf. [23] and [20] for a recent overview. Let $K_{jj}^X(t,s) = \text{Cov}(X_{ij}(t), X_{ij}(s))$ denote the covariance function for $X_{ij}$. Then, there exists orthonormal eigenfunctions and eigenvalues $\{\phi_{jk}(t), \lambda_{jk}^X\}_{k \in \mathbb{N}}$ such that for all $k \in \mathbb{N}$ [5]:

$$\int_\mathcal{T} K_{jj}^X(s,t)\phi_{jk}^X(t)dt = \lambda_{jk}^X \phi_{jk}^X(s). \quad (2.4)$$

Without loss of generality, assume $\lambda_{j1}^X \geq \lambda_{j2}^X \geq \cdots \geq 0$. By the Karhunen-Loève expansion [5, Theorem 7.3.5], $X_{ij}(t)$ can be expressed as $X_{ij}(t) = \sum_{k=1}^\infty a_{ijk}^X \phi_{jk}^X(t)$ where the principal component scores satisfy $a_{ijk}^X = \int_\mathcal{T} X_{ij}(t)\phi_{jk}^X(t)dt$ and $a_{ijk}^X \sim N(0, \lambda_{jk}^X)$ with $E(a_{ijk}^X a_{ijl}^X) = 0$ if $k \neq l$. Because the eigenfunctions are orthonormal, the $L_2$ projection of $X_{ij}$ onto the span of the first $M$ eigenfunctions is

$$X_{ij}^M(t) = \sum_{k=1}^M a_{ijk}^X \phi_{jk}^X(t). \quad (2.5)$$

Functional PCA constructs estimators $\hat{\phi}_{jk}^X(t)$ and $\hat{a}_{ijk}^X$ through the following procedure. First, we form an empirical estimate of the covariance function:

$$\hat{K}_{jj}^X(s,t) = \frac{1}{n_X} \sum_{i=1}^{n_X}(X_{ij}(s) - \bar{X}_j(s))(X_{ij}(t) - \bar{X}_j(t)),$$

where $\bar{X}_j(t) = n_X^{-1} \sum_{i=1}^{n_X} X_{ij}(t)$. An eigen-decomposition of $\hat{K}_{jj}^X(s,t)$ then directly provides the estimates $\hat{\lambda}_{jk}^X$ and $\hat{\phi}_{jk}^X$ which allow for computation of $\hat{a}_{ijk}^X = \int_\mathcal{T} X_{ij}(t)\hat{\phi}_{jk}^X(t)dt$. Let $a_{ij}^{X,M} = (a_{ij1}^X, \ldots, a_{ijM}^X)^T \in \mathbb{R}^M$ and $a_i^{X,M} = ((a_{i1}^{X,M})^T, \ldots, (a_{ip}^{X,M})^T)^T \in \mathbb{R}^{pM}$ with corresponding estimates $\hat{a}_{ij}^{X,M}$ and $\hat{a}_i^{X,M}$. Since $X_i^M(t)$ are p-dimensional MGP, $a_i^{X,M}$ will have a multivariate Gaussian distribution with $pM \times pM$ covariance matrix which we denote as $\Sigma^{X,M} = (\Theta^{X,M})^{-1}$. In practice, $M$ can be selected by cross validation as in [18].

For $(j,l) \in V^2$, let $\Theta_{jl}^{X,M}$ be the $M \times M$ matrix corresponding to $(j,l)$th submatrix of $\Theta^{X,M}$. Let $\Delta^M = \Theta^{X,M} - \Theta^{Y,M}$ be the difference between the precision matrices of the first $M$ principal component scores where $\Delta_{jl}^M$ denotes the $(j,l)$th submatrix of $\Delta^M$. In addition, let

$$E_{\Delta^M} := \{(j,l) \in V^2 : j \neq l \text{ and } \|\Delta_{jl}^M\|_F \neq 0\}, \quad (2.6)$$

denote the set of non-zero blocks of the difference matrix $\Delta^M$. In general $E_{\Delta^M} \neq E_\Delta$; however, we will see that for certain $M$, by constructing a suitable estimator of $\Delta^M$ we can still recover $E_\Delta$.

## 2.3 Functional differential graph estimation

We now describe our method, FuDGE, for functional differential graph estimation. Let $S^{X,M}$ and $S^{Y,M}$ denote the sample covariances of $\hat{a}_i^{X,M}$ and $\hat{a}_i^{Y,M}$. To estimate $\Delta^M$, we solve the following problem with the group lasso penalty, which promotes blockwise sparsity in $\hat{\Delta}^M$ [27]:

$$\hat{\Delta}^M \in \operatorname*{arg\,min}_{\Delta \in \mathbb{R}^{pM \times pM}} L(\Delta) + \lambda_n \sum_{\{i,j\} \in V^2} \|\Delta_{ij}\|_F, \tag{2.7}$$

where $L(\Delta) = \operatorname{tr}\left[\frac{1}{2} S^{Y,M} \Delta^T S^{X,M} \Delta - \Delta^T \left(S^{Y,M} - S^{X,M}\right)\right]$. Note that although the true $\Delta^M$ is symmetric, we do not enforce symmetry in $\hat{\Delta}^M$.

The design of the loss function $L(\Delta)$ in equation (2.7) is based on [15], where in order to construct a consistent M-estimator, we want the true parameter value $\Delta^M$ to minimize the population loss $\mathbb{E}\left[L(\Delta)\right]$. For a differentiable and convex loss function, this is equivalent to selecting $L$ such that $\mathbb{E}\left[\nabla L(\Delta^M)\right] = 0$. Since $\Delta^M = \left(\Sigma^{X,M}\right)^{-1} - \left(\Sigma^{Y,M}\right)^{-1}$, it satisfies $\Sigma^{X,M}\Delta^M\Sigma^{Y,M} - \left(\Sigma^{Y,M} - \Sigma^{X,M}\right) = 0$. By this observation, a choice for $\nabla L(\Delta)$ is

$$\nabla L(\Delta^M) = S^{X,M}\Delta^M S^{Y,M} - \left(S^{Y,M} - S^{X,M}\right), \tag{2.8}$$

for which $\mathbb{E}\left[\nabla L(\Delta^M)\right] = \Sigma^{X,M}\Delta^M\Sigma^{Y,M} - \left(\Sigma^{Y,M} - \Sigma^{X,M}\right) = 0$. Using properties of the differential of the trace function, this choice of $\nabla L(\Delta)$ yields $L(\Delta)$ in (2.7). The chosen loss is quadratic (see (B.10) in supplement) and leads to an efficient algorithm. Such loss has been used in [22, 26, 14] and [30].

Finally, to form $\hat{E}_\Delta$, we threshold $\hat{\Delta}^M$ by $\epsilon_n > 0$ so that:

$$\hat{E}_\Delta = \{(j,l) \in V^2 : j \neq l \text{ and } \|\hat{\Delta}_{jl}^M\|_F > \epsilon_n \text{ or } \|\hat{\Delta}_{lj}^M\|_F > \epsilon_n\}. \tag{2.9}$$

## 2.4 Optimization algorithm for FuDGE

---

**Algorithm 1** Functional differential graph estimation

---

**Input:** $S^{X,M}, S^{Y,M}, \lambda_n, \eta$.
**Output:** $\hat{\Delta}^M$.
  Initialize $\Delta^{(0)} = 0_{pM}$.
  **repeat**
    $A = \Delta - \eta\nabla L(\Delta) = \Delta - \eta\left[S_X^{(M)}\Delta S_Y^{(M)} - (S_Y^{(M)} - S_X^{(M)})\right]$
    **for** $1 \leq i,j \leq p$ **do**
      $\Delta_{jl} \leftarrow \left(\frac{\|A_{jl}\|_F - \lambda_n\eta}{\|A_{jl}\|_F}\right)_+ \cdot A_{jl}$
    **end for**
  **until** Converge

---

The optimization problem (2.7) can be solved by a proximal gradient method [17], summarized in Algorithm 1. Specifically, in each iteration step, we update the current value of $\Delta$, denoted as $\Delta^{\text{old}}$, by solving the following problem:

$$\Delta^{\text{new}} = \operatorname*{arg\,min}_{\Delta}\left(\frac{1}{2}\left\|\Delta - \left(\Delta^{\text{old}} - \eta\nabla L\left(\Delta^{\text{old}}\right)\right)\right\|_F^2 + \eta \cdot \lambda_n \sum_{j,l=1}^{p}\|\Delta_{jl}\|_F\right), \tag{2.10}$$

where $\nabla L(\Delta)$ is defined in (2.8) and $\eta$ is a user specified step size. Note that $\nabla L(\Delta)$ is Lipschitz continuous with the Lipschitz constant $\|S^{Y,M} \otimes S^{X,M}\|_2 = \lambda_{\max}(S^{Y,M})\lambda_{\max}(S^{X,M})$. Thus, for any $\eta$ such that $0 < \eta \leq 1/\lambda_{\max}^S$, the proximal gradient method is guaranteed to converge [1], where $\lambda_{\max}^S = \lambda_{\max}(S^{Y,M})\lambda_{\max}(S^{X,M})$ is the largest eigenvalue of $S^{X,M} \otimes S^{Y,M}$.

The update in (2.10) has a closed-form solution:

$$\Delta_{jl}^{\text{new}} = \left[\left(\|A_{jl}^{\text{old}}\|_F - \lambda_n\eta\right)/\|A_{jl}^{\text{old}}\|_F\right]_+ \cdot A_{jl}^{\text{old}}, \qquad 1 \leq j,l \leq p, \tag{2.11}$$

where $A^{\text{old}} = \Delta^{\text{old}} - \eta \nabla L(\Delta^{\text{old}})$ and $x_+ = \max\{0, x\}, x \in \mathbb{R}$ represents the positive part of $x$. Detailed derivations are given in the appendix.

After performing FPCA, the proximal gradient descent method converges in $O\left(\lambda_{\max}^S / \text{tol}\right)$ iterations, where tol is error tolerance, each iteration takes $O((pM)^3)$ operations. See [19] for convergence analysis of proximal gradient descent algorithm.

## 3 Theoretical properties

In this section, we present theoretical properties of the proposed method. Again, we state assumptions explicitly for $X_i(t)$, but also require the same conditions on $Y_i(t)$.

**Assumption 3.1.** *Recall that $\lambda_{jk}^X$ and $\phi_{jk}^X(t)$ are the the eigenvalues and eigenfunctions of $K_{jj}^X(t)$, the covariance function for $X_{ij}(t)$, and $\lambda_{jk}^X > \lambda_{jk'}^X$ for all $k' > k$.*

*(i) Assume $\max_{j \in V} \sum_{k=1}^{\infty} \lambda_{jk}^X < \infty$ and there exists some constant $\beta_X > 1$ such that for each $k \in \mathbb{N}$, $\lambda_{jk}^X \asymp k^{-\beta_X}$ and $d_{jk}^X \lambda_{jk}^X = O(k)$ uniformly in $j \in V$, where $d_{jk}^X = 2\sqrt{2} \max\left\{ \left( \lambda_{j(k-1)}^X - \lambda_{jk}^X \right)^{-1}, \left( \lambda_{jk}^X - \lambda_{j(k+1)}^X \right)^{-1} \right\}$.*

*(ii) Assume for all $k \in \mathbb{N}$, $\phi_{jk}^X(t)$'s are continuous on the compact set $\mathcal{T}$ and satisfy $\max_{j \in V} \sup_{s \in \mathcal{T}} \sup_{k \geq 1} |\phi_{jk}^X(s)| = O(1)$.*

The parameter $\beta_X$ controls the decay rate of the eigenvalues and $d_{jk}^X \lambda_{jk}^X = O(k)$ controls the decay rate of eigen-gaps (see [2] for more details).

To recover the exact functional differential graph structure, we need further assumptions on the difference operator $C^\Delta = \{C_{jl}^X(s,t) - C_{jl}^Y(s,t)\}_{j,l \in V}$. Let $\nu = \nu(M) = \max_{(j,l) \in V^2} \left| \|C_{jl}^\Delta\|_{\text{HS}} - \|\Delta_{jl}^M\|_F \right|$, and let $\tau = \min_{(j,l) \in E_\Delta} \|C_{jl}^\Delta\|_{\text{HS}}$, where $\tau > 0$ by the definition in (2.3). Roughly speaking, $\nu(M)$ measures the bias due to using an $M$-dimensional approximation, and $\tau$ measures the strength of signal in the differential graph. A smaller $\tau$ implies that the graph is harder to recover, and in Theorem 3.1, we require the bias to be small compared to the signal.

**Assumption 3.2.** *Assume that $\lim_{M \to \infty} \nu(M) = 0$.*

We also require Assumption 3.3 which assumes sparsity in $E_\Delta$. Again, this does not preclude the case where $E_X$ and $E_Y$ are dense, as long as the difference between the two graphs is sparse. This assumption is common in the scalar setting; e.g., Condition 1 in [30].

**Assumption 3.3.** *There are $s$ edges in the differential graph; i.e., $|E_\Delta| = s$.*

Before we give conditions for recovering the differential graph with high probability, we first introduce some additional notation. Let $n = \min\{n_X, n_Y\}$, $\sigma_{\max} = \max\{|\Sigma^{X,M}|_\infty, |\Sigma^{Y,M}|_\infty\}$, $\beta = \min\{\beta_X, \beta_Y\}$, and $\lambda_{\min}^* = \lambda_{\min}\left(\Sigma^{X,M}\right) \times \lambda_{\min}\left(\Sigma^{Y,M}\right)$. Given positive constant $c_1$, denote

$$\delta = (1/\sqrt{c_1}) M^{1+\beta} \sqrt{2\left(\log p + \log M + \log n\right)/n} \tag{3.1}$$

and

$$\Gamma = \frac{9\lambda_n^2 s}{\kappa_{\mathcal{L}}^2} + \frac{2\lambda_n}{\kappa_{\mathcal{L}}}(\omega_{\mathcal{L}}^2 + 2p^2\nu), \tag{3.2}$$

where

$$\begin{aligned}
\lambda_n &= 2M\left[\left(\delta^2 + 2\delta\sigma_{max}\right)\left|\Delta^M\right|_1 + 2\delta\right], \\
\kappa_{\mathcal{L}} &= (1/2)\lambda_{\min}^* - 8M^2 s\left(\delta^2 + 2\delta\sigma_{\max}\right), \text{ and} \\
\omega_{\mathcal{L}} &= 4Mp^2\nu\sqrt{\delta^2 + 2\delta\sigma_{\max}}.
\end{aligned} \tag{3.3}$$

Note that $\Gamma$ implicitly depends on $n$ through $\lambda_n, \kappa_{\mathcal{L}}, \omega_{\mathcal{L}}$ and $\delta$.

**Theoreom 3.1.** *There exist positive constants $c_1$ and $c_2$, such that for $n$ and $M$ large enough to simultaneously satisfy*

$$0 < \Gamma < (1/2)\tau - \nu(M) \text{ and}$$

$$\delta < \min\left\{(1/4)\sqrt{(\lambda_{\min}^* + 16M^2 s(\sigma_{\max})^2)/(M^2 s)} - \sigma_{\max}, \ c_1\right\}, \tag{3.4}$$

*setting* $\epsilon_n \in (\Gamma + \nu(M), \tau - (\Gamma + \nu(M)))$ *ensures that*

$$P\left(\hat{E}_\Delta = E_\Delta\right) \geq 1 - 2c_2/n^2.$$

[18] assumed for some finite $M$, for all $j \in V$, $\lambda_{jm'}^X = 0$ for all $m' > M$. Under this assumption, $X_{ij}(t) = X_{ij}^M(t)$, and $E_X$ will correspond exactly to $(j, l) \in V^2$ such that $\|\Theta_{jl}^{X,M}\|_F \neq 0$ [18, Lemma 1]. If the same eigenvalue condition holds for $Y_i(t)$, then in our setting $E_\Delta = E_{\Delta^M}$. When this holds and we can fix $M$, we obtain consistency even in the high-dimensional setting since $\nu = 0$ and $\min\{s \log(pn)|\Delta^M|_1^2/n, s\sqrt{\log(pn)/n}\} \to 0$ implies consistent estimation. However, even with an infinite number of positive eigenvalues, high-dimensional consistency is still possible for quickly decaying $\nu$; e.g, if $\nu = o(p^{-2}M^{-1})$ the same rate is achievable as when $v(M) = 0$.

# 4 Experiments

## 4.1 Simulation study

In this section, we demonstrate properties of our method through simulations. In each setting, we generate $n_X \times p$ functional variables from graph $G_X$ via $X_{ij}(t) = b(t)^T \delta_{ij}^X$, where $b(t)$ is a five dimensional basis with disjoint support over $[0, 1]$ with

$$b_k(t) = \begin{cases} \cos\left(10\pi\left(x - (2k-1)/10\right)\right) + 1 & (k-1)/5 \leq x < k/5; \\ 0 & \text{otherwise}, \end{cases} \qquad k = 1, \ldots, 5.$$

$\delta_i^X = ((\delta_{i1}^X)^T, \cdots, (\delta_{ip}^X)^T)^T \in \mathbb{R}^{5p}$ follows a multivariate Gaussian distribution with precision matrix $\Omega^X$. $Y_{ij}(t)$ was generated in a similar way with precision matrix $\Omega^Y$. We consider three models with different graph structures, and for each model, data are generated with $n_X = n_Y = 100$ and $p = 30, 60, 90, 120$. We repeat this 30 times for each $p$ and model setting.

**Model 1:** This model is similar to the setting considered in [30], but modified to the functional case. We generate support of $\Omega^X$ according to a graph with $p(p-1)/10$ edges and a power-law degree distribution with an expected power parameter of 2. Although the graph is sparse with only 20% of all possible edges present, the power-law structure mimics certain real-world graphs [16] by creating hub nodes with large degree. For each nonzero block, $\Omega_{jl}^X = \delta' I_5$, where $\delta'$ is sampled uniformly from $\pm[0.2, 0.5]$. To ensure positive definiteness, we further scale each off-diagonal block by $1/2, 1/3, 1/4, 1/5$ for $p = 30, 60, 90, 120$ respectively. Each diagonal element of $\Omega^X$ is set to 1 and the matrix is symmetrized by averaging it with its transpose. To get $\Omega^Y$, we first select the largest hub nodes in $G_X$ (i.e., the nodes with largest degree), and for each hub node we select the top (by magnitude) 20% of edges. For each selected edge, we set $\Omega_{jl}^Y = \Omega_{jl}^X + W$ where $W_{km} = 0$ for $|k - m| \leq 2$, and $W_{km} = c$ otherwise, where $c$ is generated in the same way as $\delta'$. For all other blocks, $\Omega_{jl}^Y = \Omega_{jl}^X$.

**Model 2:** We first generate a tridiagonal block matrix $\Omega_X^*$ with $\Omega_{X,jj}^* = I_5$, $\Omega_{X,j,j+1}^* = \Omega_{X,j+1,j}^* = 0.6I_5$, and $\Omega_{X,j,j+2}^* = \Omega_{X,j+2,j}^* = 0.4I_5$ for $j = 1, \ldots, p$. All other blocks are set to 0. We then set $\Omega_{Y,j,j+3}^* = \Omega_{Y,j+3,j}^* = W$ for $j = 1, 2, 3, 4$, and let $\Omega_{Y,jl}^* = \Omega_{X,jl}^*$ for all other blocks. Thus, we form $G_Y$ by adding four edges to $G_X$. We let $W_{km} = 0$ when $|k - m| \leq 1$, and $W_{km} = c$ otherwise, with $c = 1/10$ for $p = 30$, $c = 1/15$ for $p = 60$, $c = 1/20$ for $p = 90$, and $c = 1/25$ for $p = 120$. Finally, we set $\Omega^X = \Omega_X^* + \delta I$, $\Omega^Y = \Omega_Y^* + \delta I$, where $\delta = \max\{|\min(\lambda_{\min}(\Omega_X^*), 0)|, |\min(\lambda_{\min}(\Omega_Y^*), 0)|\} + 0.05$.

**Model 3:** We generate $\Omega_X^*$ according to an Erdös-Rényi graph. We first set $\Omega_{X,jj}^* = I_5$. With probability .8, we set $\Omega_{X,jl}^* = \Omega_{X,lj}^* = 0.1I_5$, and set it to 0 otherwise. Thus, we expect 80% of all possible edges to be present. Then, we form $G_Y$ by randomly adding $s$ new edges to $G_X$, where $s = 3$ for $p = 30$, $s = 4$ for $p = 60$, $s = 5$ for $p = 90$, and $s = 6$ for $p = 120$. We set each corresponding block $\Omega_{Y,jl}^* = W$, where $W_{km} = 0$ when $|k - m| \leq 1$ and $W_{km} = c$ otherwise. We let $c = 2/5$ for $p = 30$, $c = 4/15$ for $p = 60$, $c = 1/5$ for $p = 90$, and $c = 4/25$ for $p = 120$. Finally, we set $\Omega^X = \Omega_X^* + \delta I$, $\Omega^Y = \Omega_Y^* + \delta I$, where $\delta > \max\{|\min(\lambda_{\min}(\Omega_X^*), 0)|, |\min(\lambda_{\min}(\Omega_Y^*), 0)|\} + 0.05$.

Although the theory assumes fully observed functional data, in order to mimic a realistic setting, we use noisy observations at discrete time points, such that the actual data corresponding to $X_{ij}$ are

$$h_{ijk}^X = X_{ij}(t_k) + e_{ijk}, \quad e_{ijk} \sim N(0, 0.5^2),$$

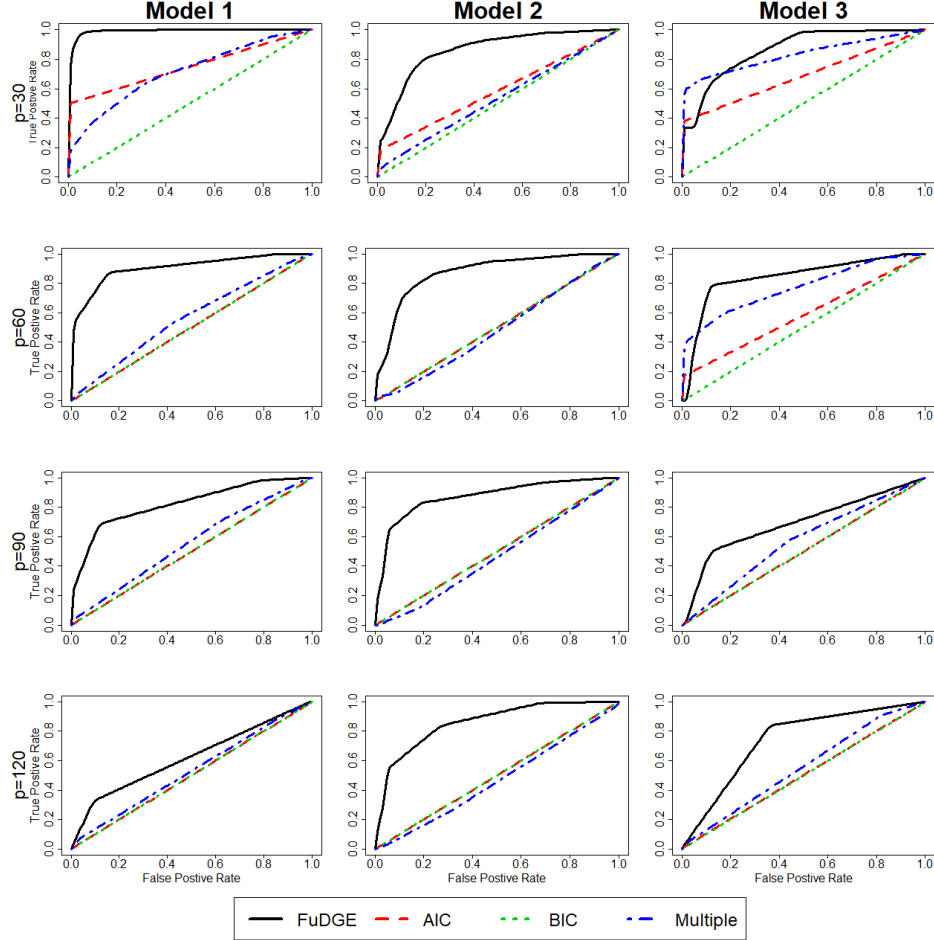

Figure 1: Average ROC curves across 30 simulations. Different columns correspond to different models, different rows correspond to different dimensions.

for 200 evenly spaced time points $0 = t_1 \le \cdots \le t_{200} = 1$. $h^Y_{ijk}$ are obtained in a similar way. For each observation, we first estimate a function by fitting an $L$-dimensional B-spline basis. We then use these estimated functions for FPCA and our direct estimation procedure. Both $M$ and $L$ are chosen by 5-fold cross-validation as discussed in [18]. Since $\epsilon_n$ in (2.9) is usually very small in practice, we simply let $\hat{E}_\Delta = \{(j, l) \in V^2 : j \ne l \text{ and } \|\hat{\Delta}^M_{jl}\|_F + \|\hat{\Delta}^M_{lj}\|_F > 0\}$. We can form a receiver operating characteristics (ROC) curve for recovery of $E_\Delta$ by using different values of the group lasso penalty $\lambda_n$ defined in (2.7).

We compare FuDGE to three competing methods. The first two competing methods separately estimate two functional graphical models using fglasso from [18]. Specifically, we use fglasso to estimate $\hat{\Theta}^{X,M}$ and $\hat{\Theta}^{Y,M}$. We then set $\hat{E}_\Delta$ to be all edges $(j, l) \in V^2$ such that $\|\hat{\Theta}^{X,M}_{jl} - \hat{\Theta}^{Y,M}_{jl}\|_F > \zeta$. For each separate fglasso problem, the penalization parameter is selected by maximizing AIC in first competing method and maximizing BIC in second competing method. We define the degrees of freedom for both AIC and BIC to be the number of edges included in the graph times $M^2$. We form an ROC curve by using different values of $\zeta$.

The third competing method ignores the functional nature of the data. We select 15 equally spaced time points and implement a direct estimation method at each time point. Specifically, for each $t$, $X_i(t)$ and $Y_i(t)$ are simply $p$-dimensional random vectors, and we use their sample covariances in (2.7) to obtain a $p \times p$ matrix $\hat{\Delta}$. This produces 15 differential graphs, and we use a majority vote to form a single differential graph. The ROC curve is obtained by changing $\lambda_n$, the $L_1$ penalty used for all time points.

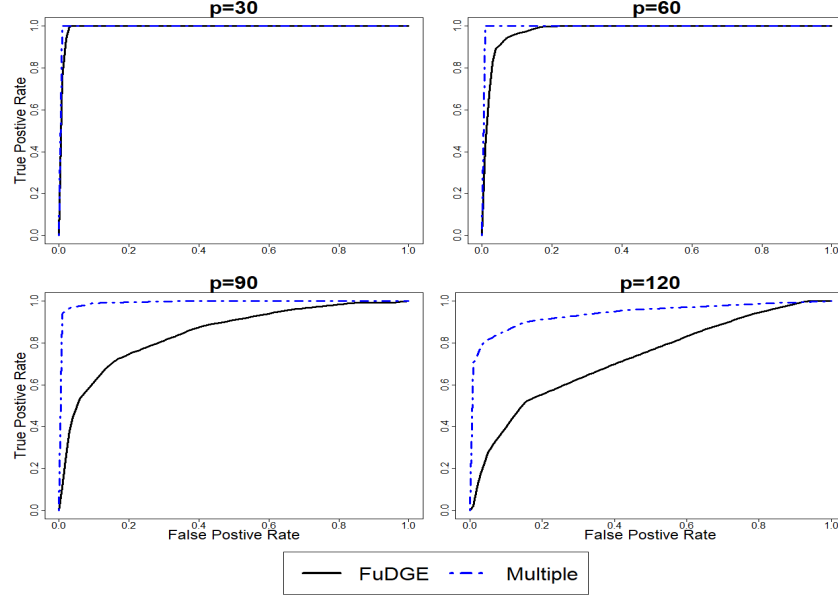

Figure 2: Average ROC curves across 30 simulations of example that multiple network strategy works better

For each setting and method, the ROC curve averaged across the 30 replications is shown in Figure 1. We see that FuDGE clearly has the best overall performance in recovering the support of differential graph. Among the competing methods, ignoring the functional structure and using a majority vote generally performs better than separately estimating two functional graphs. A table with the average area under the ROC curve is given in the appendix.

## 4.2 Example that combination of multiple networks at discrete time points works better

By construction, the simulations presented in Section 4.1 are estimating $E_\Delta$ defined in (2.3), which is not equivalent to

$$\tilde{E}_\Delta(t) = \left\{ (j,l) \in V^2 : j \neq l, |\tilde{C}_{jl}^X(t) - \tilde{C}_{jl}^Y(t)| \neq 0 \right\}, \tag{4.1}$$

where

$$\tilde{C}_{jl}^X(t) = \text{Cov}\left(X_{ij}(t), X_{il}(t) \mid \{X_{ik}(t)\}_{k \neq j,l}\right), \tag{4.2}$$

and $\tilde{C}_{jl}^Y(t)$ defined similarly. However, when $\tilde{E}_\Delta(t) = E_\Delta, \forall t$, then the differential structure can be recovered by considering individual time points. Since considering time points individually requires estimating fewer parameters than the functional version, the multiple networks strategy performs better than FuDGE.

Here, data are generated with $n_X = n_Y = 100$, and $p = 30, 60, 90, 120$. We repeat the simulation 30 times for each $p$. The model setting here is similar to model 2 in Section 4.1. However, we make two major changes. First, when we generate the functional variables, we use a 5-dimensional Fourier basis, so that all basis are supported over the entire interval, rather than disjoint support as in Section 4.1. Second, we set matrix $W$ to be diagonal. Specifically, we let $W_{kk} = c$ for $k = 1, 2, \cdots, 5$ and $W_{km} = 0$ for $k \neq m$, where $c$ is drawn uniformly from $[0.6, 1]$, and scaled by $1/2$ for $p = 30$, $1/3$ for $p = 60$, and $1/4$ for $p = 90$. All other settings are the same. The average ROC curves are shown in Figure 2, and the mean area under the curves are shown in Table 2 in section D.2 of supplementary material.

In Section 4.1 we considered extreme settings where the data must be treated as functions, and here we consider an extreme setting where the functional nature is irrelevant. In practice, however, the data may often lie between these two settings, and the method which performs better should depend on the variation of the differential structure across time. However, as it may be hard to measure this variation in practice, treating the data as functional objects should be a more robust choice.

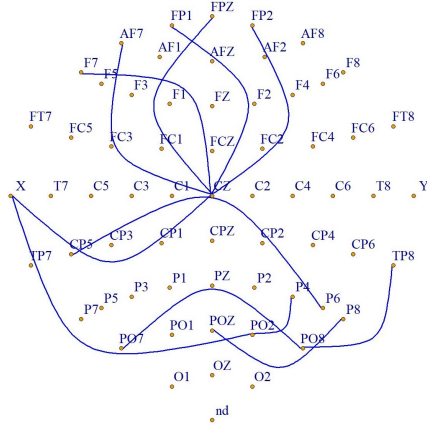

Figure 3: Estimated differential graph for EEG data. The anterior region is the top of the figure and the posterior region is the bottom of the figure.

### 4.3 Neuroscience application

We apply our method to electroencephalogram (EEG) data obtained from an alcoholism study [29, 6, 18] which included 122 total subjects; 77 in an alcoholic group and 45 in the control group. Specifically, the EEG data was measured by placing $p = 64$ electrodes on various locations on the subject's scalp and measuring voltage values across time. We follow the preprocessing procedure in [8, 31], which filters the EEG signals at $\alpha$ frequency bands between 8 and 12.5 Hz.

[18] separately estimate functional graphs for both groups, but we directly estimate the differential graph using FuDGE. We choose $\lambda_n$ so that the estimated differential graph has approximately 1% of possible edges. The estimated edges of the differential graph are shown in Figure 3.

We see that edges are generally between nodes located in the same region–either the anterior region or the posterior region–and there is no edge that crosses between regions. This observation is consistent with the result in [18] where there are no connections between frontal and back regions for both groups. We also note that electrode CZ, lying in the central region has a high degree in the estimated differential graph. While there is no direct connection between anterior and posterior regions, the central region may play a role in helping the two parts communicate.

## 5 Discussion

In this paper, we propose a method to directly estimate the differential graph for functional graphical models. In certain settings, direct estimation allows for the differential graph to be recovered consistently, even if each underlying graph cannot be consistently recovered. Experiments on simulated data also show that preserving the functional nature of the data rather than treating the data as multivariate scalars can also result in better estimation of the difference graph.

A key step in the procedure is first representing the functions with an $M$-dimensional basis using FPCA, and Assumption 3.2 ensures that there exists some $M$ large enough so that the signal, $\tau$, is larger than the bias due to using a finite dimensional representation, $\nu$. Intuitively, $\nu$ is tied to the eigenvalue decay rate; however, we defer derivation of the explicit connection for future work. Finally, we have provided a method for direct estimation of the differential graph, but development of methods which allow for inference and hypothesis testing in functional differential graphs would be fruitful avenues for future work. For example, [7] has developed inference tools for high-dimensional Markov networks, future works may extend their results to functional graph setting.

## Footnotes

[1]The code for this part is on https://github.com/boxinz17/FuDGE

[2]Both $X_i(t)$ and $Y_i(t)$ are indexed by $i$, but they are not paired observations and are completely independent. Also, we assume mean zero and a common domain $\mathcal{T}$ to simplify the notation, but the methodology and theory generalize to non-zero means and different time domains $\mathcal{T}_X$ and $\mathcal{T}_Y$ when fixing some bijection $\mathcal{T}_X \mapsto \mathcal{T}_Y$.

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
