[Supplementary Material]

# Supplementary Material for Direct Estimation of Differential Functional Graphical Models

**Boxin Zhao**
Department of Statistics
The Unveristy of Chicago
Chicago, IL 60637
boxinz@uchicago.edu

**Y. Samuel Wang**
Booth School of Business
The Unveristy of Chicago
Chicago, IL 60637
swang24@uchicago.edu

**Mladen Kolar**
Booth School of Business
The Unveristy of Chicago
Chicago, IL 60637
mkolar@chicagobooth.edu

## A  Derivation of optimization algorithm

In this section we derive the closed-form updates for the proximal method stated in (2.11). In particular, recall that for all $1 \leq j, l \leq p$

$$\Delta_{jl}^{\text{new}} = \left[\left(\|A_{jl}^{\text{old}}\|_F - \lambda_n \eta\right) / \|A_{jl}^{\text{old}}\|_F\right]_+ \times A_{jl}^{\text{old}},$$

where $A^{\text{old}} = \Delta^{\text{old}} - \eta \nabla L(\Delta^{\text{old}})$ and $x_+ = \max\{0, x\}, x \in \mathbb{R}$ represents the positive part of $x$.

*Proof of* (2.11). Let $A^{\text{old}} = \Delta^{\text{old}} - \eta \nabla L(\Delta^{\text{old}})$, and let $f_{jl}$ denote the loss decomposed over each $j, l$ block so that

$$f_{jl}(\Delta_{jl}) = \frac{1}{2\lambda_n \eta}\|\Delta_{jl} - A_{jl}^{\text{old}}\|_F^2 + \|\Delta_{jl}\|_F, \tag{A.1}$$

and

$$\Delta_{jl}^{\text{new}} = \underset{\Delta_{jl} \in \mathbb{R}^{M \times M}}{\arg\min} f_{jl}(\Delta_{jl}). \tag{A.2}$$

The loss $f_{jl}(\Delta_{jl})$ is convex, so the first order optimality condition implies that:

$$0 \in \partial f_{jl}\left(\Delta_{jl}^{\text{new}}\right), \tag{A.3}$$

where $\partial f_{jl}\left(\Delta_{jl}\right)$ is the subdifferential of $f_{jl}$ at $\Delta_{jl}$. Note that $\partial f_{jl}\left(\Delta_{jl}\right)$ can be expressed as:

$$\partial f_{jl}(\Delta_{jl}) = \frac{1}{\lambda_n \eta}\left(\Delta_{jl} - A_{jl}^{\text{old}}\right) + Z_{jl}, \tag{A.4}$$

where

$$Z_{jl} = \begin{cases} \frac{\Delta_{jl}}{\|\Delta_{jl}\|_F} & \text{if } \Delta_{jl} \neq 0 \\ \left\{Z_{jl} \in \mathbb{R}^{M \times M} : \|Z_{jl}\|_F \leq 1\right\} & \text{if } \Delta_{jl} = 0. \end{cases} \tag{A.5}$$

**Claim 1** If $\|A_{jl}^{\text{old}}\|_F > \lambda_n \eta > 0$, then $\Delta_{jl}^{\text{new}} \neq 0$.

We verify this claim by proving the contrapositive. Suppose $\Delta_{jl}^{\text{new}} = 0$, then by (A.3) and (A.5), there exists a $Z_{jl} \in \mathbb{R}^{M \times M}$ such that $\|Z_{jl}\|_F \leq 1$ and

$$0 = -\frac{1}{\lambda_n \eta}A_{jl}^{\text{old}} + Z_{jl}.$$

Thus,

$$\|A_{jl}^{\text{old}}\|_F = \|\lambda_n \eta \cdot Z_{jl}\|_F \leq \lambda_n \eta,$$

so that Claim 1 holds.

Combining Claim 1 with (A.3) and (A.5), for any $j, l$ such that $\|A_{jl}^{\text{old}}\|_F > \lambda_n \eta$, we have

$$0 = \frac{1}{\lambda_n \eta} \left( \Delta_{jl}^{\text{new}} - A_{jl}^{\text{old}} \right) + \frac{\Delta_{jl}^{\text{new}}}{\|\Delta_{jl}^{\text{new}}\|_F},$$

which is solved by

$$\Delta_{jl}^{\text{new}} = \frac{\|A_{jl}^{\text{old}}\|_F - \lambda_n \eta}{\|A_{jl}^{\text{old}}\|_F} A_{jl}^{\text{old}}. \tag{A.6}$$

**Claim 2** If $\|A_{jl}^{\text{old}}\|_F \leq \lambda_n \eta$, then $\Delta_{jl}^{\text{new}} = 0$.

Again, we verify the claim by proving the contrapositive. Suppose $\Delta_{jl}^{\text{new}} \neq 0$, then first order optimality implies the updates in (A.6). However, taking the Frobenius norm of both sides of the equation gives $\|\Delta_{jl}^{\text{new}}\|_F = \|A_{jl}^{\text{old}}\|_F - \lambda_n \eta$ which implies that $\|A_{jl}^{\text{old}}\|_F - \lambda_n \eta \geq 0$.

The updates in (2.11) immediately follow from combining Claim 2 and (A.6). $\qquad\square$

# B Proof of theoretical properties

We provide the proof of Theorem 3.1, which states that under certain conditions, our estimator consistently recovers $E_\Delta$. We follow the framework introduced in Negahban et al. (2012), but first introduce some necessary notation.

We use $\otimes$ to denote the Kronecker product. For $\Delta \in \mathbb{R}^{pM \times pM}$, let $\theta = \text{vec}(\Delta) \in \mathbb{R}^{p^2 M^2}$ and $\theta^* = \text{vec}(\Delta^M)$, where $\Delta^M$ is defined in Section 2.2. Let $\mathcal{G} = \{G_t\}_{t=1,\dots,N_\mathcal{G}}$ be a set of indices, where $N_\mathcal{G} = p^2$ and $G_t \subset \{1, 2, \cdots, p^2 M^2\}$ is the set of indices for $\theta$ which correspond to the $t$-th $M \times M$ submatrix of $\Delta^M$. Thus, if $t = (j-1)p + l$, then $\theta_{G_t} = \text{vec}(\Delta_{jl}) \in \mathbb{R}^{M^2}$ where $\Delta_{jl}$ is the $(j, l)$-th $M \times M$ submatrix of $\Delta$. Denote the group indices of $\theta^*$ that belong to blocks corresponding to $E_\Delta$ as $S_\mathcal{G} \subseteq \{1, 2, \cdots, N_\mathcal{G}\}$. Note that we define $S_\mathcal{G}$ using $E_\Delta$ and not $E_{\Delta^M}$, so as stated in Assumption 3.3, $|S_\mathcal{G}| = s$. We further define the subspace $\mathcal{M}$ as

$$\mathcal{M} := \{\theta \in \mathbb{R}^{p^2 M^2} | \theta_{G_t} = 0 \text{ for all } t \notin S_\mathcal{G}\}, \tag{B.1}$$

and its orthogonal complement with respect to the usual Euclidean inner product is

$$\mathcal{M}^\perp := \{\theta \in \mathbb{R}^{p^2 M^2} | \theta_{G_t} = 0 \text{ for all } t \in S_\mathcal{G}\}. \tag{B.2}$$

For a vector $\theta$, let $\theta_\mathcal{M}$ and $\theta_{\mathcal{M}^\perp}$ be the projection of $\theta$ on the subspaces $\mathcal{M}$ and $\mathcal{M}^\perp$, respectively. Let $\langle \cdot, \cdot \rangle$ represent the usual Euclidean inner product. Let

$$\mathcal{R}(\theta) := \sum_{t=1}^{N_\mathcal{G}} |\theta_{G_t}|_2 \triangleq \|\theta\|_{1,2}. \tag{B.3}$$

For any $v \in \mathbb{R}^{p^2 M^2}$, the dual norm of $\mathcal{R}$ is given by

$$\mathcal{R}^*(v) := \sup_{u \in \mathbb{R}^{p^2 M^2} \setminus \{0\}} \frac{\langle u, v \rangle}{\mathcal{R}(u)} = \sup_{\mathcal{R}(u) \leq 1} \langle u, v \rangle, \tag{B.4}$$

and the subspace compatibility constant of $\mathcal{M}$ with respect to $\mathcal{R}$ is defined as

$$\Psi(\mathcal{M}) := \sup_{u \in \mathcal{M} \setminus \{0\}} \frac{\mathcal{R}(u)}{|u|_2}. \tag{B.5}$$

## B.1 Proof of theoreom 3.1

Let $\sigma_{max} = \max\{|\Sigma^{X,M}|_\infty, |\Sigma^{Y,M}|_\infty\}$. Suppose that

$$\begin{aligned} |S^{X,M} - \Sigma^{X,M}|_\infty &\leq \delta, \\ |S^{Y,M} - \Sigma^{Y,M}|_\infty &\leq \delta, \end{aligned} \tag{B.6}$$

for some appropriate choice of $\delta$. Then

$$|(S^{Y,M} \otimes S^{X,M}) - (\Sigma^{Y,M} \otimes \Sigma^{X,M})|_\infty \leq \delta^2 + 2\delta\sigma_{max}, \tag{B.7}$$

and

$$|\text{vec}(S^{Y,M} - S^{X,M}) - \text{vec}(\Sigma^{Y,M} - \Sigma^{X,M})|_\infty \leq 2\delta. \tag{B.8}$$

Because by assumption $\lim_{M \to \infty} \nu(M) = 0$, there exists some $M$ large enough so that $2\nu(M) < \tau$, for $\tau$ defined in Assumption 3.2. In particular, we suppose for such $M$, that $\delta < \frac{1}{4}\sqrt{\frac{\lambda^*_{min} + 16M^2 s(\sigma_{max})^2}{M^2 s}} - \sigma_{max}$. Later, we show using Lemma C.2 that this occurs with high probability for large $n$.

Problem (2.7) can be written in following form:

$$\hat{\theta}_{\lambda_n} \in \underset{\theta \in \mathbb{R}^{p^2 M^2}}{\arg\min} \mathcal{L}(\theta) + \lambda_n \mathcal{R}(\theta), \tag{B.9}$$

where

$$\mathcal{L}(\theta) = \frac{1}{2}\theta^T(S^{Y,M} \otimes S^{X,M})\theta - \theta^T \text{vec}(S^{Y,M} - S^{X,M}). \tag{B.10}$$

The loss $\mathcal{L}(\theta)$ is convex and differentiable with respect to $\theta$, and it can be easily verified that $\mathcal{R}(\cdot)$ defines a vector norm. For $h \in \mathbb{R}^{p^2 M^2}$, the error of the first-order Taylor series expansion of $\mathcal{L}$ is:

$$
\begin{aligned}
\delta\mathcal{L}(h, \theta^*) &:= \mathcal{L}(\theta^* + h) - \mathcal{L}(\theta^*) - \langle \nabla\mathcal{L}(\theta^*), h \rangle \\
&= \frac{1}{2} h^T (S^{Y,M} \otimes S^{X,M}) h.
\end{aligned}
\tag{B.11}
$$

Using the form of (B.10), we see that $\nabla\mathcal{L}(\theta) = (S^{Y,M} \otimes S^{X,M})\theta - \text{vec}(S^{Y,M} - S^{X,M})$, and by Lemma C.1, we have

$$
\mathcal{R}^*(\nabla\mathcal{L}(\theta^*)) = \max_{t=1,2,\cdots,N_\mathcal{G}} \left| \left[ (S^{Y,M} \otimes S^{X,M})\theta^* - \text{vec}(S^{Y,M} - S^{X,M}) \right]_{G_t} \right|_2.
\tag{B.12}
$$

We now show an upper bound for $\mathcal{R}^*(\nabla\mathcal{L}(\theta^*))$. First, note that

$$
(\Sigma^{Y,M} \otimes \Sigma^{X,M})\theta^* - \text{vec}(\Sigma^{Y,M} - \Sigma^{X,M}) = \text{vec}(\Sigma^{X,M}\Delta^M\Sigma^{Y,M} - (\Sigma^{Y,M} - \Sigma^{X,M})) = 0.
$$

Letting $(\cdot)_{jl}$ denote the $(j, l)$-th submatrix, we have

$$
\begin{aligned}
&\left| \left[ (S^{Y,M} \otimes S^{X,M})\theta^* - \text{vec}(S^{Y,M} - S^{X,M}) \right]_{G_t} \right|_2 \\
&= \left| \left[ (S^{Y,M} \otimes S^{X,M} - \Sigma^{Y,M} \otimes \Sigma^{X,M})\theta^* - \text{vec}\left( (S^{Y,M} - \Sigma^{Y,M}) - (S^{X,M} - \Sigma^{X,M}) \right) \right]_{G_t} \right|_2 \\
&\leq \| (S^{X,M}\Delta^M S^{Y,M} - \Sigma^{X,M}\Delta^M\Sigma^{Y,M})_{jl} - (S^{Y,M} - \Sigma^{Y,M})_{jl} - (S^{X,M} - \Sigma^{X,M})_{jl} \|_F \\
&\leq \| (S^{X,M}\Delta^M S^{Y,M} - \Sigma^{X,M}\Delta^M\Sigma^{Y,M})_{jl} \|_F + \| (S^{Y,M} - \Sigma^{Y,M})_{jl} \|_F + \| (S^{X,M} - \Sigma^{X,M})_{jl} \|_F.
\end{aligned}
\tag{B.13}
$$

For any $M \times M$ matrix $A$, $\|A\|_F \leq M|A|_\infty$, so

$$
\begin{aligned}
&\left| \left[ (S^{Y,M} \otimes S^{X,M})\theta^* - \text{vec}(S^{Y,M} - S^{X,M}) \right]_{G_t} \right|_2 \\
&\leq M \left[ \left| (S^{X,M}\Delta^M S^{Y,M} - \Sigma^{X,M}\Delta^M\Sigma^{Y,M})_{jl} \right|_\infty + \left| (S^{Y,M} - \Sigma^{Y,M})_{jl} \right|_\infty \right. \\
&\quad \left. + \left| (S^{X,M} - \Sigma^{X,M})_{jl} \right|_\infty \right] \\
&\leq M \left[ \left| S^{X,M}\Delta^M S^{Y,M} - \Sigma^{X,M}\Delta^M\Sigma^{Y,M} \right|_\infty + \left| S^{Y,M} - \Sigma^{Y,M} \right|_\infty + \left| S^{X,M} - \Sigma^{X,M} \right|_\infty \right].
\end{aligned}
$$

Now, note that for any $A \in \mathbb{R}^{k \times k}$ and $v \in \mathbb{R}^k$, we have $|Av|_\infty \leq |A|_\infty |v|_1$, thus we further have

$$
\begin{aligned}
\left| S^{X,M}\Delta^M S^{Y,M} - \Sigma^{X,M}\Delta^M\Sigma^{Y,M} \right|_\infty &= \left| \left[ (S^{Y,M} \otimes S^{X,M}) - (\Sigma^{X,M} \otimes \Sigma^{Y,M}) \right] \text{vec}\left( \Delta^M \right) \right|_\infty \\
&\leq \left| (S^{Y,M} \otimes S^{X,M}) - (\Sigma^{X,M} \otimes \Sigma^{Y,M}) \right|_\infty \left| \text{vec}\left( \Delta^M \right) \right|_1 \\
&= \left| (S^{Y,M} \otimes S^{X,M}) - (\Sigma^{X,M} \otimes \Sigma^{Y,M}) \right|_\infty |\Delta^M|_1.
\end{aligned}
$$

Combining the inequalities gives an upper bound uniform over $\mathcal{G}$ (i.e., for all $G_t$):

$$
\begin{aligned}
&\left| \left[ (S^{Y,M} \otimes S^{X,M})\theta^* - \text{vec}(S^{Y,M} - S^{X,M}) \right]_{G_t} \right|_2 \\
&\leq M \left[ \left| (S^{Y,M} \otimes S^{X,M}) - (\Sigma^{X,M} \otimes \Sigma^{Y,M}) \right|_\infty |\Delta^M|_1 + \left| S^{Y,M} - \Sigma^{Y,M} \right|_\infty \right. \\
&\quad \left. + \left| S^{X,M} - \Sigma^{X,M} \right|_\infty \right],
\end{aligned}
$$

which implies

$$
\begin{aligned}
\mathcal{R}^*\left( \nabla\mathcal{L}(\theta^*) \right) &\leq M \left[ \left| (S^{Y,M} \otimes S^{X,M}) - (\Sigma^{X,M} \otimes \Sigma^{Y,M}) \right|_\infty |\Delta^M|_1 + \left| S^{Y,M} - \Sigma^{Y,M} \right|_\infty \right. \\
&\quad \left. + \left| S^{X,M} - \Sigma^{X,M} \right|_\infty \right].
\end{aligned}
\tag{B.14}
$$

Assuming $|S^{X,M} - \Sigma^{X,M}|_\infty \leq \delta$ and $|S^{Y,M} - \Sigma^{Y,M}|_\infty \leq \delta$ implies

$$
\mathcal{R}^*\left( \nabla\mathcal{L}(\theta^*) \right) \leq M[(\delta^2 + 2\delta\sigma_{max})|\Delta^M|_1 + 2\delta],
\tag{B.15}
$$

where $0 < \delta \le c_1$.

Setting

$$\lambda_n = 2M \left[ \left( \delta^2 + 2\delta\sigma_{max} \right) \left| \Delta^M \right|_1 + 2\delta \right],$$
(B.16)

then implies that $\lambda_n \ge 2\mathcal{R}^* \left( \nabla \mathcal{L}(\theta^*) \right)$. Thus, invoking Lemma 1 in Negahban et al. (2012), $h = \hat{\theta}_{\lambda_n} - \theta^*$ must satisfy

$$\mathcal{R}(h_{\mathcal{M}^\perp}) \le 3\mathcal{R}(h_{\mathcal{M}}) + 4\mathcal{R}(\theta^*_{\mathcal{M}^\perp}),$$
(B.17)

where $\mathcal{M}$ is defined in (B.1). Equivalently,

$$\|h_{\mathcal{M}^\perp}\|_{1,2} \le 3\|h_{\mathcal{M}}\|_{1,2} + 4\|\theta^*_{\mathcal{M}^\perp}\|_{1,2}.$$
(B.18)

By the definition of $\nu$ in Assumption 3.2, we have

$$\|\theta^*_{\mathcal{M}^\perp}\|_{1,2} = \sum_{t \notin \mathcal{S}_\mathcal{G}} \|\theta^*_{G_t}\|_2 \le (p(p+1)/2 - s)\nu \le p^2\nu.$$
(B.19)

Next, we show that $\delta\mathcal{L}(h, \theta^*)$, as defined in (B.11), satisfies the Restricted Strong Convexity property defined in definition 2 in Negahban et al. (2012). That is, we show an inequality of the form: $\delta\mathcal{L}(h, \theta^*) \ge \kappa_\mathcal{L}|h|_2^2 - \omega_\mathcal{L}^2(\theta^*)$ whenever $h$ satisfies (B.18).

By using Lemma C.3, we have

$$
\begin{aligned}
\theta^T(S^{Y,M} \otimes S^{X,M})\theta &= \theta^T(\Sigma^{Y,M} \otimes \Sigma^{X,M})\theta + \theta^T(S^{Y,M} \otimes S^{X,M} - \Sigma^{Y,M} \otimes \Sigma^{X,M})\theta \\
&\ge \theta^T(\Sigma^{Y,M} \otimes \Sigma^{X,M})\theta - |\theta^T(S^{Y,M} \otimes S^{X,M} - \Sigma^{Y,M} \otimes \Sigma^{X,M})\theta| \\
&\ge \lambda^*_{min}|\theta|_2^2 - M^2|S^{Y,M} \otimes S^{X,M} - \Sigma^{Y,M} \otimes \Sigma^{X,M}|_\infty \|\theta\|_{1,2}^2,
\end{aligned}
$$

where the last inequality holds because Lemma C.3 and $\lambda^*_{min} = \lambda_{min}(\Sigma^{X,M}) \times \lambda_{min}(\Sigma^{Y,M}) = \lambda_{min}(\Sigma^{Y,M} \otimes \Sigma^{X,M}) > 0$. Thus,

$$
\begin{aligned}
\delta\mathcal{L}(h, \theta^*) &= \frac{1}{2}h^T(S^{Y,M} \otimes S^{X,M})h \\
&\ge \frac{1}{2}\lambda^*_{min}|h|_2^2 - \frac{1}{2}M^2|S^{Y,M} \otimes S^{X,M} - \Sigma^{Y,M} \otimes \Sigma^{X,M}|_\infty \|h\|_{1,2}^2.
\end{aligned}
$$

By Lemma C.4 and (B.18), we have

$$
\begin{aligned}
\|h\|_{1,2}^2 &= (\|h_{\mathcal{M}}\|_{1,2} + \|h_{\mathcal{M}^\perp}\|_{1,2})^2 \\
&\le 16(\|h_{\mathcal{M}}\|_{1,2} + \|\theta^*_{\mathcal{M}^\perp}\|_{1,2})^2 \\
&\le 16(\sqrt{s}\|h\|_2 + p^2\nu)^2 \\
&\le 32s\|h\|_2^2 + 32p^2\nu.
\end{aligned}
$$

Combining with the equation above, we get

$$
\begin{aligned}
\delta\mathcal{L}(h, \theta^*) &\ge \left[ \frac{1}{2}\lambda^*_{\min} - 16M^2 s |S^{Y,M} \otimes S^{X,M} - \Sigma^{Y,M} \otimes \Sigma^{X,M}|_\infty \right] |h|_2^2 \\
&\quad - 16M^2 p^4 \nu^2 |S^{Y,M} \otimes S^{X,M} - \Sigma^{Y,M} \otimes \Sigma^{X,M}|_\infty \\
&\ge \left[ \frac{1}{2}\lambda^*_{\min} - 8M^2 s \left( \delta_1\delta_2 + \delta_2\sigma_{\max} + \delta_1\sigma^Y_{\max} \right) \right] |h|_2^2 \\
&\quad - 16M^2 p^4 \nu^2 \left( \delta_1\delta_2 + \delta_2\sigma_{\max} + \delta_1\sigma^Y_{\max} \right).
\end{aligned}
$$
(B.20)

Thus, appealing to (B.7), the Restricted Strong Convexity property holds with

$$\begin{aligned} \kappa_{\mathcal{L}} &= \frac{1}{2}\lambda^*_{min} - 8M^2 s\left(\delta^2 + 2\delta\sigma_{max}\right), \\ \omega_{\mathcal{L}} &= 4Mp^2\nu\sqrt{\delta^2 + 2\delta\sigma_{max}}. \end{aligned} \tag{B.21}$$

When $\delta < \frac{1}{4}\sqrt{\frac{\lambda^*_{min} + 16M^2 s(\sigma_{max})^2}{M^2 s}} - \sigma_{max}$ then $\kappa_{\mathcal{L}} > 0$. By Theorem 1 of Negahban et al. (2012) and Lemma C.4, letting $\lambda_n = 2M\left[\left(\delta^2 + 2\delta\sigma_{max}\right)|\Delta^M|_1 + 2\delta\right]$, as in (B.16), ensures

$$\begin{aligned} \|\hat{\Delta}^M - \Delta^M\|_F^2 = \|\hat{\theta}_{\lambda_n} - \theta^*\|_2^2 &\leq 9\frac{\lambda_n^2}{\kappa_{\mathcal{L}}^2}\Psi^2(\mathcal{M}) + \frac{\lambda_n}{\kappa_{\mathcal{L}}}\left(2\omega_{\mathcal{L}}^2 + 4\mathcal{R}(\theta^*_{\mathcal{M}^\perp})\right) \\ &= \frac{9\lambda_n^2 s}{\kappa_{\mathcal{L}}^2} + \frac{2\lambda_n}{\kappa_{\mathcal{L}}}(\omega_{\mathcal{L}}^2 + 2p^2\nu) \\ &\coloneqq \Gamma. \end{aligned} \tag{B.22}$$

Note that $\Gamma$ is function of $\delta$ through $\lambda_n$ (defined in (B.16)), $\kappa_{\mathcal{L}}$, and $\omega_{\mathcal{L}}$. For fixed $M$, $\nu(M)$ and $p$, $k \to 0$ as $\delta \to 0$, so there exists a $\delta_0 > 0$ such that $\delta < \delta_0$ implies

$$\begin{aligned} &\Gamma < (1/2)\tau - \nu, \\ &\delta < \min\left\{\frac{1}{4}\sqrt{\frac{\lambda^*_{min} + 16M^2 s(\sigma_{max})^2}{M^2 s}} - \sigma_{max},\ c_1\right\}, \end{aligned} \tag{B.23}$$

for any $c_1 > 0$. When these hold, there exists an

$$\epsilon_n \in (\Gamma + \nu, \tau - (\Gamma + \nu)), \tag{B.24}$$

and when thresholding with this $\epsilon_n$ we claim $\hat{E}_{\Delta^M} = E_\Delta$. We prove this claim below.

Note that we have $\|\hat{\Delta}_{jl} - \Delta_{jl}^M\|_F \leq \|\hat{\Delta} - \Delta^M\|_F \leq \Gamma$ for any $(j, l) \in V^2$. Recall that

$$E_\Delta = \{(j, l) \in V^2 : \|C_{jl}^\Delta\|_{\mathrm{HS}} > 0, j \neq l\}. \tag{B.25}$$

We first prove that $E_\Delta \subseteq \hat{E}_{\Delta^M}$. For any $(j, l) \in E_\Delta$, by the definition of $\nu$ and $\tau$ in Assumption 3.2, we have $\|C_{jl}^\Delta\|_{\mathrm{HS}} \geq \tau$ and $\|\Delta_{jl}^M\|_F \geq \|C_{jl}^\Delta\|_{\mathrm{HS}} - \nu$. Thus, we have

$$\begin{aligned} \|\hat{\Delta}_{jl}\|_F &\geq \|\Delta_{jl}^M\|_F - \|\hat{\Delta}_{jl} - \Delta_{jl}^M\|_F \\ &\geq \|C_{jl}^\Delta\|_{\mathrm{HS}} - \|\hat{\Delta}_{jl} - \Delta_{jl}^M\|_F - \nu \\ &\geq \tau - \Gamma - \nu \\ &> \epsilon_n. \end{aligned}$$

The last inequality holds because we have assumed that $\epsilon_n \in (\Gamma + \nu(M), \tau - (\Gamma + \nu(M)))$. Thus, by definition of $\hat{E}_{\Delta^M}$ shown in (2.9), we have $(j, l) \in \hat{E}_{\Delta^M}$ which further implies that $E_\Delta \subseteq \hat{E}_{\Delta^M}$.

We then show $\hat{E}_{\Delta^M} \subseteq E_\Delta$. Let $\hat{E}_{\Delta^M}^c$ and $E_\Delta^c$ denote the complement set of $\hat{E}_{\Delta^M}$ and $E_\Delta$. For any $(j, l) \in E_\Delta^c$, which also means that $(l, j) \in E_\Delta^c$, by (B.25), we have $\|C_{jl}^\Delta\|_{\mathrm{HS}} = 0$, thus

$$\begin{aligned} \|\hat{\Delta}_{jl}\|_F &\leq \|\Delta_{jl}^M\|_F + \|\hat{\Delta}_{jl} - \Delta_{jl}^M\|_F \\ &\leq \|C_{jl}^\Delta\|_{\mathrm{HS}} + \|\hat{\Delta}_{jl} - \Delta_{jl}^M\|_F + \nu \\ &\leq \Gamma + \nu \\ &< \epsilon_n. \end{aligned}$$

Again, the last inequality holds because because we have assumed that $\epsilon_n$ satisfies (B.24). Thus, by definition of $\hat{E}_{\Delta^M}$, we have $(j, l) \notin \hat{E}_{\Delta^M}$ or $(j, l) \in \hat{E}_{\Delta^M}^c$. This implies that $E_\Delta^c \subseteq \hat{E}_{\Delta^M}^c$, or $\hat{E}_{\Delta^M} \subseteq E_\Delta$. Combing with previous conclusion that $E_\Delta \subseteq \hat{E}_{\Delta^M}$, the proof is complete.

We now show that for any $\delta$, there exists some $n$ large enough so that, (B.6), (B.7) and (B.8) occur with high probability. In particular, let

$$\delta \;=\; \frac{1}{\sqrt{c_1}} M^{1+\beta_X} \sqrt{\frac{2\,(\log p + \log M + \log n)}{n}}, \tag{B.26}$$

where $\lim_{n \to \infty} \delta(n) = 0$. Thus, there exists some $n$ large enough such that $\delta_0 = \delta(n)$ satisfies (B.23). Then, Lemma C.2 implies that there exists some $c_1$, $c_2$ such that (B.6), (B.7) and (B.8) holds for $\delta < c_1$ with probability $1 - 2c_2/n^2$.

# C   Lemmas in the proof of theoretical properties

**Lemma C.1.** *For $\mathcal{R}(\cdot)$ norm defined in (B.3), its dual norm $\mathcal{R}^*(\cdot)$, defined in (B.4), is*

$$\mathcal{R}^*(v) = \max_{t=1,\dots,N_{\mathcal{G}}} |v_{G_t}|_2. \tag{C.1}$$

*Proof.* For any $u : \|u\|_{1,2} \leq 1$ and $v \in \mathbb{R}^{p^2 M^2}$, we have

$$
\begin{aligned}
\langle v, u \rangle &= \sum_{t=1}^{N_{\mathcal{G}}} \langle v_{G_t}, u_{G_t} \rangle \\
&\leq \sum_{t=1}^{N_{\mathcal{G}}} |v_{G_t}|_2 |u_{G_t}|_2 \\
&\leq \left( \max_{t=1,2,\cdots,N_{\mathcal{G}}} |v_{G_t}|_2 \right) \sum_{t=1}^{N_{\mathcal{G}}} |u_{G_t}|_2 \\
&= \left( \max_{t=1,2,\cdots,N_{\mathcal{G}}} |v_{G_t}|_2 \right) \|u\|_{1,2} \\
&\leq \max_{t=1,2,\cdots,N_{\mathcal{G}}} |v_{G_t}|_2.
\end{aligned}
$$

To complete the proof, we to show that this upper bound can be obtained. Let $t^* = \arg\max_{t=1,2,\cdots,N_{\mathcal{G}}} |v_{G_t}|$, and select $u$ such that

$$
\begin{aligned}
u_{G_t} &= 0 & \forall t \neq t^*, \\
u_{G_t} &= \frac{v_{G_{t^*}}}{|v_{G_{t^*}}|_2} & t = t^*.
\end{aligned}
$$

It follows that $\|u\|_{1,2} = 1$ and $\langle v, u \rangle = |v_{G_{t^*}}|_2 = \max_{t=1,\dots,N_{\mathcal{G}}} |v_{G_t}|_2$. $\qquad\square$

**Lemma C.2.** *Let*

$$f(n, p, M, \delta, \beta, c_1, c_2) = c_2 p^2 M^2 \exp\left\{ -c_1 n M^{-(2+2\beta)} \delta^2 \right\}, \tag{C.2}$$

$\beta = \min\{\beta_X, \beta_Y\}$ *where $\beta_X$ and $\beta_Y$ are as defined in Assumption 3.1, and $\sigma_{max} = \max\{\sigma_{max}^X, \sigma_{max}^Y\}$ where $\sigma_{max}^X$ and $\sigma_{max}^Y$ are as defined in Section 3.*

*There exists positive constants, $c_1$ and $c_2$, such that for $0 < \delta < c_1$, with probability at least $1 - 2f(\min\{n_X, n_Y\}, p, M, \delta, \beta, c_1, c_2)$ the following statements hold simultaneously:*

$$
\begin{aligned}
|S^{X,M} - \Sigma^{X,M}|_\infty &\leq \delta, \\
|S^{Y,M} - \Sigma^{Y,M}|_\infty &\leq \delta,
\end{aligned} \tag{C.3}
$$

$$|(S^{Y,M} \otimes S^{X,M}) - (\Sigma^{Y,M} \otimes \Sigma^{X,M})|_\infty \leq \delta^2 + 2\delta\sigma_{max}, \tag{C.4}$$

*and*

$$|\operatorname{vec}(S^{Y,M} - S^{X,M}) - \operatorname{vec}(\Sigma^{Y,M} - \Sigma^{X,M})|_\infty \leq 2\delta. \tag{C.5}$$

*Proof.* Denote the $(j, l)$-th $M \times M$ submatrix of $S^{X,M}$ by $S_{jl}^{X,M}$ and the $(k, m)$-th entry of $S_{jl}^{X,M}$ by $\hat{\sigma}_{jl,km}^{X,M}$ for $j, l = 1, \dots, p$ and $k, m = 1, \dots, M$. We use similar notation for $\Sigma^{X,M}$, $S^{Y,M}$, and $\Sigma^{Y,M}$.

The statement in (C.3) holds directly by applying Theorem 1 in Qiao et al. (2019) to $S^{X,M}$ and $S^{Y,M}$ and combining the statements with a union bound.

To show (C.4), note that (C.3) then implies

$$
\begin{aligned}
|\hat{\sigma}^{X,M}_{jl,km}\hat{\sigma}^{Y,M}_{j'l',k'm'} - \Sigma^{X,M}_{jl,km}\sigma^{Y,M}_{j'l',k'm'}| &\leq |\hat{\sigma}^{X,M}_{jl,km} - \sigma^{X,M}_{jl,km}||\hat{\sigma}^{Y,M}_{j'l',k'm'} - \sigma^{Y,M}_{j'l',k'm'}| \\
&\quad + |\hat{\sigma}^{X,M}_{jl,km}||\hat{\sigma}^{Y,M}_{j'l',k'm'} - \sigma^{Y,M}_{j'l',k'm'}| \\
&\quad + |\hat{\sigma}^{Y,M}_{j'l',k'm'}||\hat{\sigma}^{X,M}_{jl,km} - \sigma^{X,M}_{jl,km}| \\
&\leq |S^{X,M} - \Sigma^{X,M}|_\infty |S^{Y,M} - \Sigma^{Y,M}|_\infty \\
&\quad + \sigma_{max}|S^{Y,M} - \Sigma^{Y,M}|_\infty + \sigma_{max}|S^{X,M} - \Sigma^{X,M}| \\
&\leq \delta^2 + 2\delta\sigma_{max}.
\end{aligned}
$$

For (C.5), note that

$$
\begin{aligned}
|\operatorname{vec}\left(S^{Y,M} - S^{X,M}\right) - \operatorname{vec}\left(\Sigma^{Y,M} - \Sigma^{X,M}\right)|_\infty &= |(S^{X,M} - \Sigma^{X,M}) - (S^{Y,M} - \Sigma^{Y,M})|_\infty \\
&\leq |S^{X,M} - \Sigma^{X,M}|_\infty + |S^{Y,M} - \Sigma^{Y,M}|_\infty \\
&\leq 2\delta.
\end{aligned}
$$

$\square$

**Lemma C.3.** *For a set of indices $\mathcal{G} = \{G_t\}_{t=1,\dots,N_\mathcal{G}}$, suppose $\|\cdot\|_{1,2}$ is defined in (B.3). Then for any matrix $A \in \mathbb{R}^{p^2 M^2 \times p^2 M^2}$ and $\theta \in \mathbb{R}^{p^2 M^2}$*

$$
|\theta^T A\theta| \leq M^2 |A|_\infty \|\theta\|^2_{1,2}. \tag{C.6}
$$

*Proof.*

$$
\begin{aligned}
|\theta^T A\theta| &= \left|\sum_i \sum_j A_{ij}\theta_i\theta_j\right| \\
&\leq \sum_i \sum_j |A_{ij}\theta_i\theta_j| \\
&\leq |A|_\infty \left(\sum_i |\theta_i|\right)^2 \\
&= |A|_\infty \left(\sum_{t=1}^{N_\mathcal{G}} \sum_{k\in G_t} |\theta_k|\right)^2 \\
&= |A|_\infty \left(\sum_{t=1}^{N_\mathcal{G}} \|\theta_{G_t}\|_1\right)^2 \\
&\leq |A|_\infty \left(\sum_{t=1}^{N_\mathcal{G}} M\|\theta_{G_t}\|_2\right)^2 \\
&= M^2 |A|_\infty \|\theta\|^2_{1,2}.
\end{aligned}
$$

In the penultimate line, we use the property that for any vector $v \in \mathbb{R}^n$, $|v|_1 \leq \sqrt{n}|v|_2$. $\square$

**Lemma C.4.** *Suppose $\mathcal{M}$ is defined as in (B.1). For any $\theta \in \mathcal{M}$, we have $\|\theta\|_{1,2} \leq \sqrt{s}|\theta|_2$. Furthermore, for $\Psi(\mathcal{M})$ as defined in (B.5), we have $\Psi(\mathcal{M}) = \sqrt{s}$.*

*Proof.* By definition of $\mathcal{M}$ and $\|\cdot\|_{1,2}$, we have

$$\|\theta\|_{1,2} = \sum_{t \in S_\mathcal{G}} |\theta_{G_t}|_2 + \sum_{t \notin S_\mathcal{G}} |\theta_{G_t}|_2$$

$$= \sum_{t \in S_\mathcal{G}} |\theta_{G_t}|_2$$

$$\leq \sqrt{s} \left( \sum_{t \in S_\mathcal{G}} |\theta_{G_t}|_2^2 \right)^{\frac{1}{2}}$$

$$= \sqrt{s} |\theta|_2.$$

In the penultimate line, we appeal to the Cauchy-Schwartz inequality. To show $\Psi(\mathcal{M}) = \sqrt{s}$, it suffices to show that the upper bound above can be achieved. Select $\theta \in \mathbb{R}^{p^2 M^2}$ such that $|\theta_{G_t}|_2 = c$, $\forall t \in S_\mathcal{G}$, where $c$ is some positive constant. This implies that $\|\theta\|_{1,2} = sc$ and $|\theta|_2 = \sqrt{s}c$ so that $\|\theta\|_{1,2} = \sqrt{s}|\theta|_2$. Thus, $\Psi(\mathcal{M}) = \sqrt{s}$.

$\square$

# D More simulation results

## D.1 AUC table of simulations in section 4.1

Table 1: The mean area under the ROC curves. Standard errors are shown in parentheses.

|       | FuDGE       | AIC         | BIC     | Multiple    |
|-------|-------------|-------------|---------|-------------|
| $p$   |             | Model1      |         |             |
| 30    | 0.99 (0.01) | 0.75 (0.17) | 0.5 (0) | 0.71 (0.11) |
| 60    | 0.91 (0.06) | 0.5 (0)     | 0.5 (0) | 0.56 (0.1)  |
| 90    | 0.82 (0.1)  | 0.5(0)      | 0.5 (0) | 0.55 (0.09) |
| 120   | 0.64 (0.06) | 0.5(0)      | 0.5 (0) | 0.53 (0.04) |
| $p$   |             | Model2      |         |             |
| 30    | 0.9 (0.08)  | 0.59 (0.06) | 0.5 (0) | 0.53 (0.14) |
| 60    | 0.9 (0.07)  | 0.5 (0)     | 0.5 (0) | 0.48 (0.11) |
| 90    | 0.88 (0.08) | 0.5(0)      | 0.5 (0) | 0.46 (0.08) |
| 120   | 0.86 (0.07) | 0.5(0)      | 0.5 (0) | 0.46 (0.12) |
| $p$   |             | Model3      |         |             |
| 30    | 0.87 (0.06) | 0.69 (0.06) | 0.5 (0) | 0.83 (0.08) |
| 60    | 0.83 (0.09) | 0.58 (0.07) | 0.5 (0) | 0.77 (0.09) |
| 90    | 0.74 (0.1)  | 0.5(0)      | 0.5 (0) | 0.57 (0.1)  |
| 120   | 0.74 (0.08) | 0.5(0.02)   | 0.5 (0) | 0.55 (0.05) |

## D.2 AUC table of simulations in section 4.2

Table 2: The mean area under the ROC curves of example that multiple network strategy works better. Standard errors are shown in parentheses

| $p$ | FuDGE       | Multiple    |
|-----|-------------|-------------|
| 30  | 0.99 (0)    | 1 (0)       |
| 60  | 0.98 (0.01) | 1 (0)       |
| 90  | 0.87 (0.09) | 1 (0.01)    |
| 120 | 0.73 (0.12) | 0.94 (0.09) |