[Reviews · NeurIPS 2019]

Reviewer 1



The authors describe a method for estimating the difference between two functional graphical models using time-varying data. This is done by first modelling the functional graphical models as multi-variate Gaussian processes, and then defining the differential graph as arising from the difference between the covariance functions estimated for both processes. Optimization is done via a proximal gradient approach, and the method is evaluated under 3 different data generating mechanisms, before being applied to an EEG dataset. As I am not an expert in functional data analysis, I cannot vouch for the originality except to say that I have not come across a similar method. The quality of the method and experiments is high, and the inclusion of theoretical consistency results is welcomed. It would have been nice to have seen, in addition to the different data generating mechanisms, an exploration of the behaviour for different time series length and different numbers of variables. The latter is included in the supplementary material, but it would have been good to have at least a discussion of these results in the main text. The presentation of the manuscript was clear, and I was able to follow most of it despite it being somewhat removed from my expertise. Section 2.2. was a bit mystifying, but I suspect that is due to lack of background knowledge rather than any fault of the authors'. I would think that the significance of this method is high, as functional data is ubiquitous, and robust methods for comparing time series are hard to come by. Minor points: - The title should probably be "Direct Estimation of Differential Functional Graphical Models" rather than "Model". - In 2.1, it's not completely clear to me what makes this cross-covariance function "conditional"? I see the conditioning in the expression, but how is this different from an unconditional cross-covariance function? - Section 3, there is a typo in "eignvalues". - For the different data generation models in the simulation study, I would have liked to see an illustration of the kinds of networks that arise from these models. - It would also be useful for reproducibility if the authors could make available the code that generated the simulation data. - The authors do not seem to have plans to make the code available, which would be a shame.

Reviewer 2



1. Background and Related work are completely missing, making the paper hard to follow: 2. No clear empirical comparison and discussions connecting to the previous methods. For instance, multiple toolboxes exist from the previous differential graph models. However, the manuscripts included no results from these baselines. It seems that the computational cost is huge compared to the differential graphical models in the literature. Without a clear benefit of empirical improvements versus the tools, the significance of the paper is questionable. 3. Besides, the fPCA process is the same as the fglasso framework. The difference is the new loss formulation in this manuscript. However, the intuition or justification hasn't been discussed in the main draft about their new loss. 4. Computational Complexity wrt $p$ or $n$ hasn't been discussed. This is an extremely important perspective. However, the manuscript includes no discussions, both theoretically and experimentally. 5. In the experiments, a major concern is that the number of dimensions is low. The maximum $p$ considered is 120 and in most cases, $n$ is larger than $p$. Thus no experimental results show and justify the high dimension consistency claim. More experiments to show the bottleneck that limits $p$ to 120 or why larger $p$ haven't been considered should be added. 6. In the experiments, some guidance should be added to guide when assuming data as functional is advantageous vs estimating from discrete time points. Some related results are in the appendix, however, a clear discussion about the advantages of one method over the other hasn't been discussed. This is particularly relevant for the EEG experiment: Only qualitative results for this dataset have been added, how do the other baselines (estimating individually vs direct differential) differ in the types of edges recovered? 7. The simulation results are only on one set of data parameters, how does the underlying true sparsity of the graph affect the estimation performance? 8. Minor Comments - Experiment details are missing: How is $M$ and $L$ chosen? using cross-validation? - Some notations are confusing and should be revised: For example, $t$ is used multiple times to represent different variables, for iterations as well as time points.

Reviewer 3



The paper proposes a method to learn a differential functional graph for two multi-output functions X and Y. An edge in the graph indicates the difference of the conditional dependence of X and of Y for the pair of the nodes (i.e., variables) the edge connects. In order to do so, the paper introduces finite approximation of the random functions, where the bases are first estimated from eigen-decomposition over a kernel matrix on finite observations, and the estimates of the coefficients are obtained through integration. The coefficient estimates for each function are then used to construct a sample covariance matrix, based on which an objective function is proposed to estimate the prevision of the differential graph. The sparsity is induced from the group-lasso penalty. Several theoretical properties are given. The evaluation is conducted on simulation and neural science applications. Overall, the problem of estimating a differential functional graph sounds interesting and meaningful. However, the proposed approach seems a straightforward extension of the work by Qiao et. al. 2019 [12]. The paper lacks justification of the proposed objective. The notations and description of the method are messy. These issues hinder me from accepting this paper. Listed are detailed comments. (1) Limited novelty. The core techniques used in the work --- functional principled component analysis to obtain a finite random function approximation, using coefficients of the bases functions to build sample covariance matrices and learn the structure via a graphical lasso based optimization --- are exactly the same as the Qiao et. al. 2019 [12]. The only difference is the objective. To me, at least from the algorithmic point of view, the approach is too incremental. (2) A key component of this work is the proposed objective in eq. (2.7). However, the justification/intuition of the objective is unclear. Obviously, you cannot use graphical lasso objective. But how to justify the proposed objective? What does the first term of L(\Delta) imply? Although the authors try to give some intuition, the explanation is very vague. Why is the expectation of gradient of L zero? Why is \Delta^M the “unique” minimizer? The authors should explain them clearly, because this is the major contribution. (3) The presentation of the proposed method is really messy. The authors never differentiate scalars, vectors and matrices. The authors often confound the notations for the function X and the observation for X, say, X_i. For example, (2.1) defines the covariance between j-th function of X at s and l-th function of X at t. Obviously, the subscript “i" should be dropped. Such kinds of mixed usage of symbols are everywhere. It is really confusing to read. I have to double check [12] from time to time for confirmation.

Reviewer 4



This paper introduces an algorithm for the direct estimation of the difference between two functional graphical models. This work relies on Functional Graphical Models developed by Qiao et al. and theoretical results therein (functional PCA and convergence analysis). The paper is in general written clearly. Theorems and their proofs seem correct. The idea seems novel, at least for FGMs. However, the techniques employed to prove theorems look similar to those in Qiao et al. I do not have a specific comment about the theoretical results, but have some concerns about experiments. Qiao et al. mentioned that there is a caveat when applying AIC/BIC-based model selection methods: “However, given the complicated functional structure of FGM, it is unclear how to compute the effective degrees of freedom for AIC/BIC.“ But the paper simply chose the parameter “to be the number of edges included in the graph times M^2”. Can you please provide a reference / justification where this is a good choice? With respect to the experiment for the EEG data (Figure 2), it would be good to compare it to a result based on Qiao et al. In their paper, they presented two graphs, and their edge-differences. However, they didn’t present a result based on ||Theta^X - Theta^Y|| > a threshold. Comparing two methods based on the EEG data (with matching number of different edges) will help readers understand their differences more clearly. minor comments - Algorithm 1, Initialize \Delta^{0} can be just \Delta since no index is used throughout the algorithm. - max → \max (L143) - maximizing? or minimizing? L225

Reviewer 5



The paper seems to provide a novel and technically strong contribution, which seems to be clearly described and with a clear significance. Unfortunately, not being my field of expertise, I’m a bit at a loss about most of the technical details.

[Author Response · NeurIPS 2019]

**Novelty:** We study the problem of learning differential graphs from functional data compared to existing literature that focused on scalar data. While some of the tools we used exist in the literature, we carefully needed to improve a number of these tools to tackle challenges that arise from the infinite-dimensional functional data observed at large number of vertices. For example, while fPCA has been used in [12], they made a strong assumption that there is finite number of nonzero eigenvalues (Condition 2 in [12]), which restricts functional data to lie in a finite-dimensional space. We relax this assumption — see Assumption 3.2 and Theorem 3.1 — thus handling the truly infinite dimensional problem. Moreover, we carefully analyze the bias due to dimensionality reduction, so that we can gain consistency in a high-dimensional setting with sparsity imposed only on the differential graph, rather than separate graphs.

**Loss function:** The design of the loss function $L(\Delta)$ in equation (2.7) is based on [9], where in order to construct a consistent M-estimator, we want the true parameter value $\Delta^M$ to minimize the population loss $\mathbb{E}\left[L(\Delta)\right]$. For a differentiable and convex loss function, this is equivalent to selecting $L$ such that $\mathbb{E}\left[\nabla L(\Delta^M)\right] = 0$. Since $\Delta^M = \left(\Sigma^{X,M}\right)^{-1} - \left(\Sigma^{Y,M}\right)^{-1}$, it satisfies $\Sigma^{X,M}\Delta^M\Sigma^{Y,M} - (\Sigma^{Y,M} - \Sigma^{X,M}) = 0$. By this observation, a choice for $\nabla L(\Delta)$ is $\nabla L(\Delta) = S^{X,M}\Delta S^{Y,M} - (S^{Y,M} - S^{X,M})$ so that $\mathbb{E}\left[\nabla L(\Delta^M)\right] = \Sigma^{X,M}\Delta^M\Sigma^{Y,M} - (\Sigma^{Y,M} - \Sigma^{X,M}) = 0$, and from this choice of $\nabla L(\Delta)$, we get $L(\Delta)$ in equation (2.7) by using properties of differential of trace function. The chosen loss is quadratic (see equation (B.10) in supplement) and leads to an efficient algorithm. Such loss has been used in [15, 17], see also [21]. We will provide additional discussion in the final version.

**R2: 1.** Due to space constraints, we have only included necessary background related to presentation of our ideas. We will include additional background in the supplement of the final version. **2.** Differential graphs have been studied in a scalar setting, while there is no existing literature for the functional data setting. This makes finding a reasonable competitor difficult. Nonetheless we include two competitors. The first competitor is based on [12] — which proposes a method for estimating a single functional graph — and estimates two separate graphs before finding the differential graph. As pointed in reviews and also in [12], choosing tuning parameters for this method is challenging. Here we adopt a heuristic AIC and BIC criterion. Since each edge corresponds to an $M \times M$ block in precision matrix, the degrees of freedom is defined as the the number of edges included in the graph times $M^2$. One advantage of the FuDGE procedure is that we do not need to select multiple hyperparameters but can select a single penalty parameter for the entire problem. In addition, when both underlying graphs are dense, but the difference graph is sparse, the direct difference estimation can perform much better than the separate estimation procedure. The second competitor is to ignore the functional nature of data and directly apply scalar differential graph estimators at various fixed time points to obtain several differential graphs. The final differential graph is then an aggregate of the multiple graphs; we choose majority vote. There are at least two problems with this approach. First, in order to implement this approach, we have to have curves X and Y observed at same time points, and all sample curves need to share the same observation time grids. Both of these requirements make this competitor applicable only in simulations. Second, even if the above requirements are satisfied, there is loss of information by ignoring the functional nature of data. This is demonstrated in simulations. Thus, preserving the entire functional nature of the data is critical for graph estimation. Of course, if we know in advance that the structural information is completely captured at each time point, this procedure estimates fewer parameters and can perform better, as we considered in Supplement D. Many differential estimators can be used for estimation from one time point and here we compare against the state-of-the-art. Our main argument is to stress the strength of preserving the functional nature of the data, while different methods for scalar graphs should not make much difference. This is also related to the discussion of the novelty of the procedure in that our proposed procedure can outperform more naive procedures in the settings for which it is designed. **4.** After performing fPCA, the proximal gradient descent method converges in $O\left(L/\epsilon\right)$ iterations, where $L$ is the largest eigenvalue of $S^{X,M} \otimes S^{Y,M}$ and $\epsilon$ is error tolerance. Each iteration takes $O((pM)^3)$ operations. We will include complexity analysis in final version. **5.** We consider data of a similar size as in [21]. The differential graph we are estimating is represented by $(pM) \times (pM)$ matrix, which contains many more parameters than sample size. Moreover, we are dealing functional data, which is intrinsically an infinite-dimensional object. **6.** Please see point 2. In the final version, we will include for EEG data out of sample prediction and compare against methods that learn precision matrices separately. **7.** Note that the three models tried in the simulations actually represent very different structures in the true underlying graphs. Model 1 is a generally sparse model, but there are hub nodes, which means that some nodes can be densely connected. Model 2 is a very sparse model while model 3 is a very dense model. The differential structures are all sparse, since we are trying to demonstrate that one benefit of direct estimation is that we only require sparsity in differential graph rather than individual graphs.

**R3:** Please see the top of the response for discussion of novelty and the loss function. **3.** We have provided only the necessary background for the paper due to space constraints and we will carefully check notation for consistency in the final version.

**R1:** The code is available on github and a link will be included in the final version.

**R4:** See 2 and 6 in R2. We will include discussion of scientific findings in the final version, comparing to [12].

[Meta-Review · NeurIPS 2019]

The paper introduces a method for directly estimating the difference between two functional undirected graphical models, instead of doing it naively, and then combining them, the proposed method is novel, non-trivial, and leads to robust inferences. The authors provide extensive simulations to corroborate with their findings. Further, I like that even though some of the tools are well-studied and basic (e.g., fPCA), the authors generalized some key components in non-trivial fashion to make the whole thing to work. Having said that, and not taking any points from the technical contributions of the paper, I would be curious to see whether these new results would translate to the directed case, which is more related to causal inference. In particular, there’s a growing literature on the problem of ‘transportability” (e.g., [Bareinboim and Pearl, Proc. of Nat. Acad., of Sci, 2016]), which defines and builds exactly on a combined representation that overlaps two causal diagrams, which was called selection diagram. I wonder if the insights of this paper could be applied to learn this type of representation.